# Quantification of caffeine in coffee cans using electrochemical measurements, machine learning, and boron-doped diamond electrodes

Tatsuya Honda[1,2], Kenshin Takemura[1]*, Susumu Matsumae[2], Nobutomo Morita[1], Wataru Iwasaki[1], Ryoji Arita[1,2], Suguru Ueda[2], Yeoh Wen Liang[2], Osamu Fukuda[2], Kazuya Kikunaga[1], Shinya Ohmagari[1]

**1** Sensing System Research Center, National Institute of Advanced Industrial Science and Technology (AIST), Tosu, Saga, Japan, **2** Graduate School of Science and Engineering, Saga University, Saga, Japan

* takemura.kenshin@aist.go.jp

## Abstract

Electrochemical measurements, which exhibit high accuracy and sensitivity under low contamination, controlled electrolyte concentration, and pH conditions, have been used in determining various compounds. The electrochemical quantification capability decreases with an increase in the complexity of the measurement object. Therefore, solvent pretreatment and electrolyte addition are crucial in performing electrochemical measurements of specific compounds directly from beverages owing to the poor measurement quality caused by unspecified noise signals from foreign substances and unstable electrolyte concentrations. To prevent such signal disturbances from affecting quantitative analysis, spectral data of voltage-current values from electrochemical measurements must be used for principal component analysis (PCA). Moreover, this method enables highly accurate quantification even though numerical data alone are challenging to analyze. This study utilized boron-doped diamond (BDD) single-chip electrochemical detection to quantify caffeine content in commercial beverages without dilution. By applying PCA, we integrated electrochemical signals with known caffeine contents and subsequently utilized principal component regression to predict the caffeine content in unknown beverages. Consequently, we addressed existing research problems, such as the high quantification cost and the long measurement time required to obtain results after quantification. The average prediction accuracy was 93.8% compared to the actual content values. Electrochemical measurements are helpful in medical care and indirectly support our lives.

## Introduction

Caffeine is a naturally occurring methylated xanthine alkaloid (1,3,7-trimethylxan, 137X) that increases basal metabolism and is used as a central nervous system stimulant, myocardial stimulant, and smooth muscle relaxant [1]. In the United States, approximately 85% of the

**Data Availability Statement:** All relevant data are within the paper and its Supporting Information files.

**Funding:** This work supported following financial disclosure: " the Adaptable and Seamless Technology transfer Program through Target-driven R&D (A-STEP) from Japan Science and Technology Agency Grant Number JPMJTR22R2. The funders had no role in study design, data collection and analysis, decision to publish, or preparation of the manuscript.

**Competing interests:** The authors have declared that no competing interests exist.

population consumes caffeine daily [2]. Brewed coffee is a common source of caffeine, with 100 mg per 177 ml and 65 mg per 177 ml of instant coffee. Caffeine can also be found in cold, allergic, and headache remedies, diuretics, and stimulants [1]. Coffee consumption is associated with bone loss, decreased bone density, and fractures [3]. Pregnant women also have an increased risk of poor fetal growth and spontaneous abortion [4]. Excessive caffeine consumption has also been linked to headaches, nausea, and anxiety. In addition, the lethal dose of caffeine is estimated to be $\geq 10$ g [5]. Therefore, quantification of caffeine content is crucial for living a healthy life. Various methods have been used to estimate caffeine intake. These methods include high-performance liquid chromatography (HPLC), ultra-violet (UV) spectroscopy, thin-Layer chromatography-mass spectrometry (TLC-MS), and gas chromatography [6–10]. These methods have high accuracy despite being expensive and time consuming.

Electrochemistry has often been utilized as a sensor to measure the redox potential of substances using easily controlled voltages and currents [11]. Electrochemically sensitive caffeine measurements have been successful [12]. However, electrochemical measurements require control of the solvent pH, electrolytic mass, and other factors to ensure stable and accurate measurements [13]. Furthermore, when measuring only specific materials from a solution containing many foreign substances, the solution must be pretreated by columns and filtering before being used for measurement. Moreover, the redox voltages of the compounds that react at the electrode interface often overlap, making peak separation challenging. Boron-doped diamond (BDD) electrodes have excellent characteristics, such as a wide-potential window, low background current, and long-term response stability [14, 15]. A wide-potential window contributes to various measurable redox voltages. Furthermore, changes have been observed in the current value at which the caffeine redox reactions occur.

Machine learning enables the analysis of a large amount of data, which is a complex process for humans [16], and allows computers to discover hidden patterns and rules by providing abundant data as input. Moreover, it can provide more accurate results with higher precision [17]. Machine learning algorithms can be classified into five types: analysis, regression, clustering, dimensionality reduction, and anomaly detection [18]. Thousands of multivariate variables (solute redox reactions) are observed using electrochemical analysis. Therefore, pretreatment is crucial. However, by acquiring feature values from multiple variables using machine learning and performing dimension reduction [19] and regression analyses [20], unknown solutions can be quantified from an insignificant amount of training data. Because machine learning enables rapid analysis that considers multivariate reactions, using it in electrochemistry will be extremely effective.

In this study, electrochemical measurements of commercially available caffeine-containing beverages were performed using BDD, and the obtained results were analyzed using machine learning for highly accurate caffeine quantification. The beverages used as measurement solutions were not pretreated and contained significant amounts of foreign substances. Therefore, we realized a method to quantify the amount of caffeine using machine learning without human preprocessing.

## Methods

### Synthesis of boron-doped diamond electrode

The BDD electrode was fabricated under the conditions reported previously [21]. Heavily boron-doped polycrystalline diamond films were prepared on Si (100) substrates using hot-filament chemical vapor deposition. The substrate surfaces were treated by scare life polishing or lift-off techniques using C+ ion plating, as appropriate. The surface roughness is less than 0.1 nm in Ra. Before the film growth, the substrates were chemically cleaned at 250°C using a

mixed acid solution of $H_2SO_4$ and $HNO_3$. Before film growth, the samples were preseeded with diamond nanopowders to facilitate the nucleation of diamond on a foreign substrate. Hydrogen, methane, and trimethylboron gases were fed into the chamber at a total pressure of 1.3 kPa. The methane/hydrogen gas ratio was maintained at 3% during growth. Next, the tungsten filament wires were resistively heated by a DC power supply at a filament temperature of 2200˚C. The film thickness and boron concentration, as measured by secondary ion mass spectrometry, were 5 m and $>10^{20}$ cm$^{-3}$, respectively. The BDD electrode used was formed on a Si substrate with a film thickness of 2μm. The doped boron concentration is 5E20cm-3. The surface terminated groups are hydrogen terminated.

## Characterization of BDD

The BDD-deposited Si wafer surface was analyzed by atomic force microscopy (AFM; MFP-3D Origin+, Oxford Instruments, Abingdon-on-Thames, UK) and scanning electron microscopy (SEM; JSM-9100F, JEOL Ltd., Tokyo, Japan). Various measurements were performed using an electrochemical analyzer (ALS610C; BAS Inc., Tokyo, Japan) to confirm the electrode properties. The measurements were performed using a three-electrode system with BDD as the working electrode. An Ag/AgCl electrode was used as the reference electrode, and a Pt coil electrode was used as the counter electrode (BAS Inc., Tokyo, Japan). The electric double-layer capacitance in a 1 g/L NaCl solution was measured using cyclic voltammetry. The voltage was swept from 0 V to 0.1 V, and the scan speed was measured every 10 mV from 10 to 100 mV. Furthermore, cyclic voltammetry was performed to evaluate the stability of the electrode. The voltage was swept from -2.4 V to 2.5 V and cycled 50 times.

## Electrochemical measurement of caffeine in solution

Experiments were conducted to measure caffeine levels in commercial coffee without pretreatment. The pH levels of five different commercial coffee beverages were obtained. To verify the electrochemical response to caffeine, square wave voltammetry (SWV) measurements were performed using a 1 g/L NaCl solution containing 60 mg/100 g of anhydrous caffeine. The measurements were obtained from -2.4 V to 2.5 V in 0.016 V steps, with a 2 V voltage applied for 100 s to clean the electrode. A 3 ml solution was taken from the coffee can and directly injected into the measuring cell to perform the caffeine measurement by SWV. Measurements were taken in the laboratory, under conditions that kept the temperature at about 23˚C on average. Absorbance measurements were performed from 220 to 800 nm to check for contamination at the electrode interface using UV-2600i (Shimadzu Co. Ltd., Kyoto, Japan).

## Principal component analysis and machine learning algorism for caffeine quantification

As illustrated in Fig 1A, the electrochemically measured solution data were stored in the training data file, and the data for one of the solutions (one solution was measured three times) was stored as an unknown solution in the test data file. The solution data stored in the training data file was used to create a regression dataset of unknown solutions. The manufacturer-published value was provided as the correct solution data for each solution. The solution data stored in the test data file underwent principal component regression according to the data frame created by the data stored in the training data file. For this, principal component analysis, normalization, and grid research were performed on the solution data. Principal component analysis (PCA) was performed using Fig 1B and 1C. In this study, we developed an algorithm that combines an analytical evaluation value (the value extracted from the whole part where the feature value is strongly expressed), as shown in Fig 1B, which is the coordinate

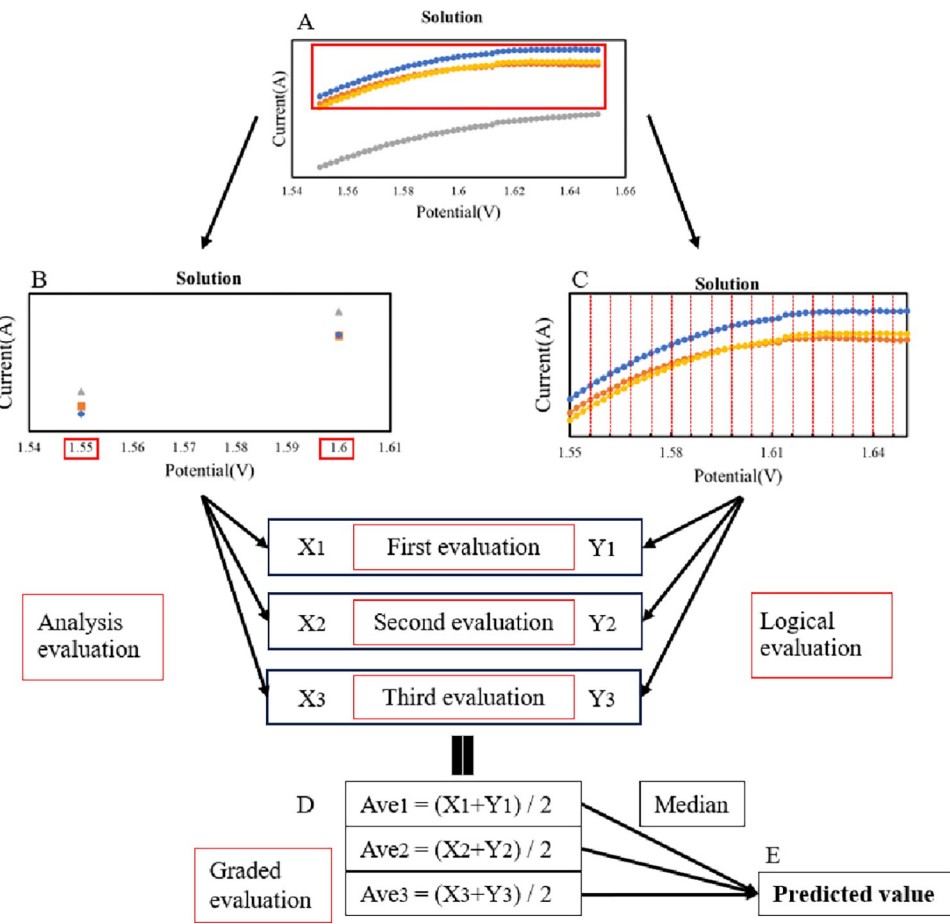

**Fig 1. Schematic diagram of the quantification algorithm for commercial beverages.**

with good results of PCA for the solution data, and a logical evaluation value (value using the area of the narrow part where the redox reaction of caffeine occurs) (Fig 1C), which is the area around the peak voltage of 1.6 V where the oxidation reaction of caffeine occurs [22]. Therefore, the data entered in the training data file creates two data sets based on these coordinates and areas. Principal component regression (PCR) [23] was performed on the solution data stored in the test data file according to the two created data sets to obtain predicted values from each. The algorithm shown in Fig 1B can obtain a value close to the caffeine content. The predicted values ($X_1$–$X_3$) were obtained using only concrete information. The algorithm shown in Fig 1C uses an area divided into 17 segments because 51 coordinates exist between 1.55 V and 1.65 V. Predictions were made using an abstract feature from the overall information (the details from the entire area over which the caffeine oxidation reaction occurred). Using this method, the predicted values ($Y_1$–$Y_3$) were obtained from the area. These predicting algorithms had low accuracy when used individually. Therefore, combining these two algorithms improves the average accuracy of the predicted values by evenly following the electrochemical measurement data, including measurement errors owing to each electrode.

Following is the explanation of the process until the data stored in the test data file are quantified. The PCA obtained in Fig 1B and 1C is used to perform the same analysis for the unknown solution. Specifically, the solution data stored in the test data file is also subjected to the principal component analysis and normalization used when creating the data set. The data

stored in the test file is adapted to the dataset. PCR was used to predict the unknown solutions from the variance obtained from the PCA results [24]. Because each solution was measured three times, three specific predictions ($X_1$–$X_3$) for solution A were obtained from Fig 1B, and three abstract predictions ($Y_1$–$Y_3$) were obtained from Fig 1C. The predicted values were averaged at each step of the hierarchy, as shown in Fig 1D. These values were considered the average values ($Ave_1$–$Ave_3$). The median of the stepwise evaluation values obtained becomes a predicted value that is finally output as a quantitative value, as shown in Fig 1E.

A more concise equation for deriving the quantitative values from the output data was calculated following equation. α, β, γ: median value μ: average value of median values.

$$\alpha = (X_1 + Y_1), \beta = (X_2 + Y_2), \gamma = (X_3 + Y_3)$$

$$\mu = \varepsilon \div 2$$

The obtained results were equivalent to this equation.

## Results and discussion

### Characterization of BDD

SEM observations of the BDD surfaces showed numerous microscale irregularities. Moreover, the deposition method was used to form thick polycrystalline diamonds on a Si surface, as shown in Fig 2A. An AFM analysis of the surface suggested that the unevenness was 100–300 nm, as shown in Fig 2B and 2C. The surface morphology of the BDD was maintained after one night of standing in 0.1 M nitric acid, as shown in S1 Fig, indicating that the BDD electrode surface has high chemical resistance. The electrochemical stability of the BDD electrodes was evaluated using continuous cyclic voltammograms in solvent-containing electrolytes, as shown in Fig 2D. The current was significantly reduced when the electrode surface was affected by the application of high voltage or by the adhesion of minute bubbles owing to water electrolysis. The electrochemical spectra showed no decrease in BDD from 0 to 1 V. Additionally, the gold electrode is a common electrode, which exhibited a continuous signal drop caused by the elution of gold during high-voltage applications, as shown in S2A Fig. GCE, composed of carbon, shows a rapid increase in the current value because the water electrolysis depends on the cycle number under the same conditions as those of BDD, as shown in S2B Fig. Therefore, the BDD electrode has a wider potential window than the commercial electrodes and can acquire stable signals even after repeated measurements at high voltages. To assess the viability of the double layer capacitance (Cdl) measurements under commercial canned coffee conditions (1 g/L salt equivalent), voltage sweeps were conducted on a BDD electrode, as shown in Fig 2E. A scan rate-dependent increase in current value was observed; to calculate Cdl, a calibration curve was constructed using the capacitance current density at the nonfaradic point (0.05 V) to determine ($|ja-jc|/2$), where $ja$ and $jc$ denote the anodic and cathodic peak current densities, respectively, as shown in Fig 2F [25]. The value of the Cdl parameter for BDD in NaCl solution was 0.18 $\mu F/cm^2$, as determined by the slope of the calibration curve. Therefore, an electric double layer was present at the interface despite being in solutions with a low electrolytic mass. Moreover, electrochemical measurements using BDD as the working electrode are viable for commercial coffee beverages despite the solvent not being pretreated. Electrodes for controlling the redox reactions of substances, mainly in commercial beverages in which significant amounts of foreign substances are present, must be highly chemically resistant. Therefore, BDD is the most suitable electrode for measuring caffeine in coffee without pretreatment.

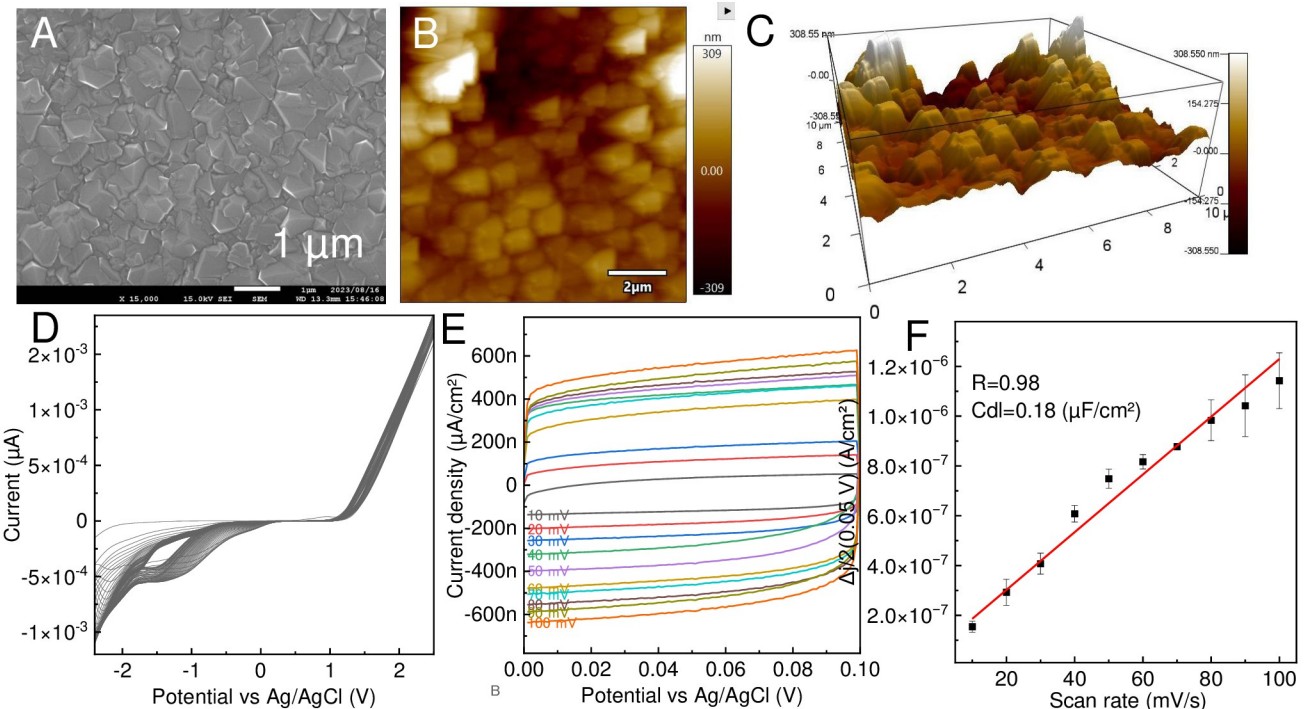

**Fig 2. A) Surface observation of BDD captured by SEM. B and C) Surface roughness of BDD analyzed by AFM. D) Electrochemical stability of BDD measured using CV. E) Cdl measurement of BDD. F) Linier plotting from CV measurement for Cdl estimation**.

**Fig 2. Property analysis of BDD.** A) Surface observation of BDD captured by SEM. B and C) Surface roughness of BDD analyzed by AFM. D) Electrochemical stability of BDD measured using CV. E) Cdl measurement of BDD. F) Linier plotting from CV measurement for Cdl estimation.

## Electrochemical measurement of caffeine

To confirm the potential at which the caffeine oxidation reaction occurred in the measurement system with the BDD electrode as the working electrode, an NaCl solution and a solution of 60 mg/100 g of caffeine suspended in NaCl were measured, as shown in Fig 3A. The addition of caffeine increased the peak current value at 1.6 V owing to caffeine oxidation. A peak attributable to the oxidation of caffeine is also observed at 0.4–0.8 V. In this potential range, many substances, such as OH groups in organic compounds, show oxidative responses. Therefore, it is difficult to differentiate the oxidation response from foreign signals in actual commercial coffee beverages. Only the 1.6 V peak is used for actual quantification. The pH and salt equivalents of the caffeine-suspended NaCl solutions and commercial beverages were summarized to evaluate the effect of pH on the measurement of commercial beverages (S1 Table). Therefore, the neutrality of both solvents did not shift the measured potential owing to the changes in pH. Commercial coffee beverages were electrochemically measured without pretreatment to confirm the response potential of caffeine, as shown in Fig 3B and 3C. The electrochemical spectra obtained showed a distinct peak at 1.6 V attributable to caffeine. In addition, a wide electrochemical peak from 0 to 1 V was observed for all the samples. The oxidation of organic compounds, such as those with hydroxyl groups, will occur in this voltage range [26, 27], indicating high amounts of foreign substances in commercial coffee beverages. When four commercial coffee beverages (Asahi, Asahi, KIRIN, and Suntory) were subjected to HPLC, 79 compounds were detected in all samples and matched to all database matches. Caffeine was found in the largest quantity despite many other organic compounds being present (S2 Table).

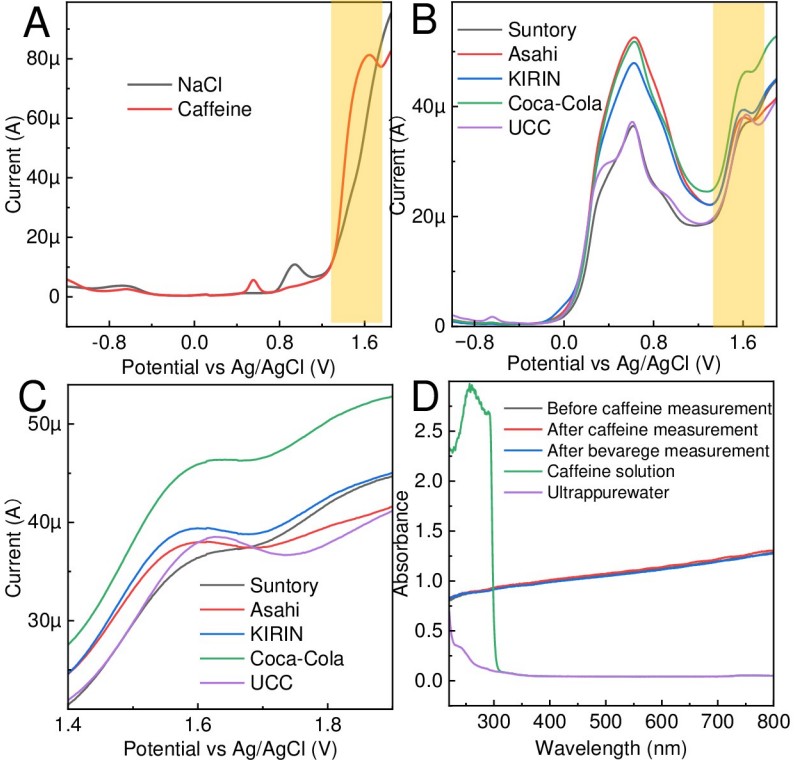

Fig 3. A) Results of SWV measure[...] NaCl solutions and caffeine suspe[...] solutions. B, C) Results of SWV measurements of commercial caff[...] beverages. D) Absorbance measu[...] BDD and caffeine solutions and u[...] water.

**Fig 3. Electrochemical caffeine measurement.** A) Results of SWV measurements of NaCl solutions and caffeine suspension solutions. B, C) Results of SWV measurements of commercial caffeine beverages. D) Absorbance measurements of BDD and caffeine solutions and ultrapure water.

The results for the caffeine peak were compared with those obtained under conditions in which only caffeine was suspended in NaCl solution, indicating that the rate of increase in the background current was influenced by the large peak at 0 V. Therefore, quantifying caffeine in commercial beverages without pretreatment using a general method of quantification using calibration curves prepared using standards is challenging. The absorbance at 272 nm, which is characteristic of caffeine, was measured to confirm caffeine adsorption at the electrode interface before and after measurement, as shown in Fig 3D. No peaks indicating caffeine adsorption were identified under either condition for BDD compared with the caffeine suspension solution and pure BDD. Therefore, caffeine was not adsorbed at the electrode interface in the direct measurement of commercial beverages, and the BDD electrode showed a certain degree of stability in repeated measurements using a single electrode. The increase in current values obtained from the direct measurement of commercial beverages was compared with the assumed content provided by the manufacturer. Furthermore, visually determining the difference in content between the types of beverages was challenging, indicating that the multiple voltage peaks resulting from high amounts of adulterants had a significant impact on the signal analysis. To quantify the caffeine content from signals obtained without pretreatment, data, and analytical techniques must be combined.

## Machine learning for quantification of caffeine in beverages

Three prediction patterns were considered in this quantification algorithm: one based on concrete information, one based on abstract information, and one combining both. The accuracy

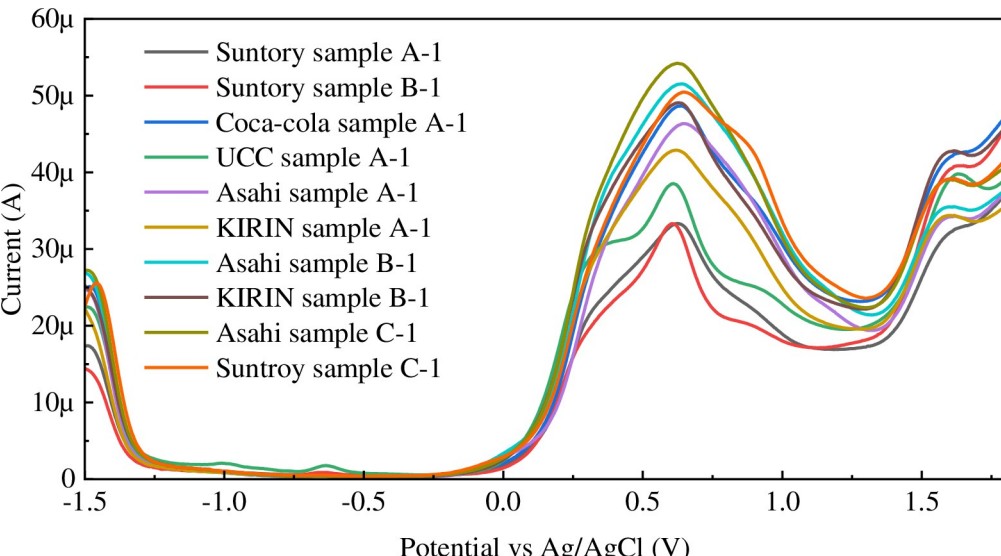

**Fig 4. Results of SWV measurements of 10 coffee samples from five companies.**

of each was examined. For concrete information, the coordinates with the highest contribution ratio and largest variance in the PCA were used. To obtain the abstract information, the areas before and after the completion of caffeine oxidation were divided into 17 parts. In this study, PCA was used because the amount of training data was insignificant in relation to the number of dimensions of the feature values. Hyperparameters were used for dimensionality reduction [28, 29].

Ten samples from five companies were assayed, including a sample of decaffeinated coffee with 60 mg of caffeine added (Fig 4). Table 1 presents the quantification results of the study. Each sample was measured three times and subjected to quantification by machine learning (S3 Fig).

Table 1 lists the algorithms that output predictions from the analysis and logical and graded evaluations. The bold values indicate the algorithm with the highest prediction accuracy among the three algorithms. It can be seen that the analysis and logical values were predicted to be close to the manufacturer's published values. However, because the algorithms calculate the predicted values either by looking only at the details or the whole, one algorithm is not superior to the other.

In addition, as listed in Table 2, the Suntory and UCC samples had recovery rates of 119% and 86.1%, respectively, which are lower than those of the other samples. These samples are Suntory sample A and UCC sample A, as listed in Table 1. In this study, the measurement was performed three times until it was stabilized (when the three measurement graphs roughly overlap) by visual observation. Therefore, if the graphs were visually stable twice, the third one is closest to it. In other words, it was considered that by recognizing something stable twice as correct, the third time would be closer to it, and the error in the quantitative would become larger if the first two times were incorrect. It is assumed that three measurements were stabilized near Suntory sample A-3 and UCC sample A-2 in Table 2. Based on these graded evaluations, the manufacturer's published values for Suntory sample A-3 and UCC sample A-2 were 40 mg (per 100 g) and 60 mg (per 100 g), respectively, while the graded evaluations were 41.86 mg (per 100 g) and 56.25 mg (per 100 g). They had the same quantitative accuracy compared to other samples. Therefore, we considered that it would be possible to further improve the

**Table 1. Caffeine quantification results for each algorithm.**

| Caffeine content (mg/100 g) | | | | |
|---|---|---|---|---|
| Unknown solution | Manufacturer published value | Analysis evaluation | Logical evaluation | Graded evaluation |
| Suntory sample A-1 | 40 | 47.46 | **47.70** | **47.58** |
| Suntory sample A-2 | | **47.30** | 49.79 | 48.55 |
| Suntory sample A-3 | | 38.97 | 44.75 | 41.86 |
| Suntory sample B-1 | 40 | **38.56** | 46.07 | 42.32 |
| Suntory sample B-2 | | 38.48 | 45.05 | 41.91 |
| Suntory sample B-3 | | 38.61 | **45.89** | **42.25** |
| Coca-Cola sample A-1 | 60 | **62.48** | **62.94** | **62.71** |
| Coca-Cola sample A-2 | | 56.87 | 54.74 | 55.80 |
| Coca-Cola sample A-3 | | 62.72 | 66.57 | 64.65 |
| UCC sample A-1 (Decaffeinated coffee) + anhydrous caffeine | 60 | **50.44** | **52.85** | **51.65** |
| UCC sample A-2 (Decaffeinated coffee) + anhydrous caffeine | | 57.14 | 55.36 | 56.25 |
| UCC sample A-3 (Decaffeinated coffee) + anhydrous caffeine | | 49.19 | 51.11 | 50.15 |
| Asahi sample A-1 | 62 | 59.03 | 54.43 | 56.73 |
| Asahi sample A-2 | | **60.89** | **60.19** | **60.54** |
| Asahi sample A-3 | | 63.70 | 65.10 | 64.40 |
| KIRIN sample A-1 | 70 | 70.00 | 76.72 | 73.36 |
| KIRIN sample A-2 | | **64.14** | **70.80** | **67.47** |
| KIRIN sample A-3 | | 61.33 | 63.04 | 62.18 |
| Asahi sample B-1 | 70 | 76.34 | 74.76 | 75.55 |
| Asahi sample B-2 | | 70.78 | **66.59** | **68.69** |
| Asahi sample B-3 | | **70.5** | 65.69 | 68.09 |
| KIRIN sample B-1 | 78 | 75.05 | 64.24 | 69.64 |
| KIRIN sample B-2 | | **75.69** | **70.79** | **73.24** |
| KIRIN sample B-3 | | 76.18 | 71.89 | 74.03 |
| Asahi sample C-1 | 80 | 80.78 | 85.60 | 83.19 |
| Asahi sample C-2 | | 83.02 | **80.42** | **81.72** |
| Asahi sample C-3 | | **82.65** | 77.42 | 80.04 |
| Suntory sample C-1 | 82 | 78.27 | **82.47** | **80.37** |
| Suntory sample C-2 | | 81.90 | 91.80 | 86.85 |
| Suntory sample C-3 | | **78.51** | 77.19 | 77.85 |

quantitative accuracy if there was a method to check whether the data used for prediction were stable.

Table 3 lists the errors between the quantified values according to Table 2 and the published values. The bold text in Table 3 (Suntory sample B, KIRIN sample A, and Suntory sample C) demonstrates that even if the quantitative values of one of the algorithms deviate significantly between the algorithm that looks at the details and the one that looks at the whole, the other is accurately quantified. This suggests that the quantitative accuracy was unstable when only one algorithm was used.

The prediction accuracy and confidence intervals of the graded evaluations were also evaluated (Table 4). This study enhanced the process using machine learning without human pre-processing. Consequently, 80% were quantified with an accuracy of 100% before and after the amount of caffeine published by the manufacturer, 95% were quantified with an accuracy of 85%, and 90% were quantified with an accuracy of 90%. The average rate was 93.88%, and the median was 95.95%.

**Table 2. Details of graded evaluation.**

| Caffeine content (mg/100 g) | | | | |
|---|---|---|---|---|
| Unknown solution | Manufacturer published value | Predicted value | Recovery (%) | Subtract the remainder of Recovery divided by 100 from 100 (%) |
| Suntory sample A | 40 | 47.58 | 119.0 | 81.0 |
| Suntory sample B | 40 | 42.25 | 105.6 | 94.4 |
| Coca-Cola sample A | 60 | 62.70 | 104.5 | 95.5 |
| UCC sample A (Decaffeinated coffee) +anhydrous caffeine | 60 | 51.64 | 86.1 | 86.1 |
| Asahi sample A | 62 | 60.54 | 97.6 | 97.6 |
| KIRIN sample A | 70 | 67.46 | 96.4 | 96.4 |
| Asahi sample B | 70 | 68.68 | 98.1 | 98.1 |
| KIRIN sample B | 78 | 73.24 | 93.9 | 93.9 |
| Asahi sample C | 80 | 81.72 | 102.2 | 97.8 |
| Suntory sample C | 82 | 80.36 | 98.0 | 98.0 |

**Table 3. Error between each predicted value and manufacturer's published value.**

| Caffeine content (mg/100 g) | | | |
|---|---|---|---|
| Unknown liquid | Analysis evaluation | Logical evaluation | Graded evaluation |
| Suntory sample A | -7.30 | -7.70 | -7.58 |
| Suntory sample B | **1.44** | **-5.89** | **-2.25** |
| Coca-Cola sample A | -2.48 | -2.94 | -2.71 |
| UCC sample A (Decaffeinated coffee) +anhydrous caffeine | 9.56 | 7.15 | 8.35 |
| Asahi sample A | 1.11 | 1.81 | 1.46 |
| KIRIN sample A | **5.86** | **-0.80** | **2.53** |
| Asahi sample B | -0.50 | 3.41 | 1.31 |
| KIRIN sample B | 2.31 | 7.21 | 4.76 |
| Asahi sample C | -2.65 | -0.42 | -1.72 |
| Suntory sample C | **3.49** | **-0.47** | **1.63** |

The principle, number of pretreatments to perform the measurement with high accuracy, and expected measurement time, including pretreatment, were compared with other studies (Table 5). Compared to standard HPLC, the number of pretreatments and measurement times were significantly higher. HPLC showed 99.2% quantitative accuracy for commercial beverages, indicating that HPLC is a simple measurement method, but its accuracy is inferior. The colorimetric and Raman spectrophotometric detection methods used a smaller number of pretreatments and measurement times to quantify commercial beverages or prepared caffeine. In both methods, a calibration curve was prepared to determine the quantity of caffeine, and it was necessary to prepare a standard for each case. The major advantage of this study is that it can be

**Table 4. Correct solution rate.**

| Contained section (%) | Content rate (%) |
|---|---|
| more than 80 | 100 |
| more than 85 | 95 |
| more than 90 | 90 |
| **Average (%)** | **Median (%)** |
| 93.88 | 95.95 |

**Table 5. Comparison with other caffeine detection in coffee beverages.**

| | means of measurement | Sample solution | Step number until measurement | Measurement time (min) | Accuracy | Ref |
|---|---|---|---|---|---|---|
| 1 | HPLC | commercial beverages | 6 | 120 | 99.2 | [31] |
| 2 | Colorimetric | commercial beverages | 2 | 15 | —* | [32] |
| 3 | Surface enhanced Raman scattering | Chemical agents | 2 | within 15 | —* | [30] |
| 4 | Electrochemistry | 0.01 M $H_2SO_4$ | 1 | within 10 | 99.9 | [33] |
| This research | Electrochemistry | commercial beverages | 0 | 2 | 93.88 | |

* Not stated because no experiments were conducted with commercial beverages.

used for product evaluation without the need to create a calibration curve if it has been studied in advance. Comparisons were made with other electrochemical caffeine sensors. By fabricating an electrode with a structure suitable for caffeine and measuring under optimal solvent conditions, we succeeded in achieving ultrahigh sensitivity and 99.9% accuracy in a single step. Electrochemical sensors can be constructed with high sensitivity under optimal conditions. In this study, no pretreatment was performed, and the electrochemical double layer formed at the electrode interface in the solution was not stable because no electrolyte was added. This leads to measurement instability and a reduced sensitivity. By utilizing machine learning, we succeeded in quantification with an accuracy of over 90% even from products containing emulsifiers without using any electrolytes, whereas it was originally necessary to establish optimal conditions as in previous studies. However, this is difficult to achieve using other electrochemical sensors.

## Conclusion

This study aimed to experimentally explore the feasibility of quantifying a manufacturer's published values without the need for added reagents. In previous studies, quantifying components dissolved in solution was expensive or time consuming [16, 30]. Quantification requires the addition of reagents to extract the caffeine spectrum. Furthermore, constructing a calibration curve requires more time and effort. Therefore, machine learning has been employed to overcome these limitations. Consequently, quantification was possible with an average accuracy of approximately 94%. However, it was crucial to reexamine the vagueness of the definition of the data, such as the current value of the measured data being roughly similar in all three cases. Furthermore, the manufacturer-published values were quantified as teacher data using machine learning with electrochemically measured data. Therefore, this technology could enhance the versatility of electrochemical measurements by improving data reliability by changing the method of scrutinizing measured data.

The combination of electrochemical measurements and machine learning demonstrated that quantitative estimation is possible even from solutions containing foreign substances, as long as the peaks are prominent. However, many organic compounds, including caffeine, have similar oxidation potentials. To make this technique more versatile, a new data learning method that mechanically processes the minor differences in oxidation potential between substances with similar oxidation potentials to separate peaks at a high level is required. In the future, we will verify whether it is possible to quantify other components with high contribution rates using a similar algorithm.

## Supporting information

**S1 Fig. SEM image of BDD after acid treatment.**
(TIF)

**S2 Fig.** A) 100 cycles of the gold electrode in NaCl solution. B) 100 cycle of glassy carbon electrode in NaCl solution.
(TIF)

**S3 Fig.** A-I) Results of 10 coffee samples from five companies measured repeatedly by SWV.
(TIF)

**S1 Table. pH measurements of beverages.**
(DOCX)

**S2 Table. HPLC measurement results of coffee beverages.**
(DOCX)

## Author Contributions

**Conceptualization:** Tatsuya Honda, Kenshin Takemura, Susumu Matsumae.

**Formal analysis:** Tatsuya Honda, Kenshin Takemura, Nobutomo Morita, Wataru Iwasaki, Ryoji Arita, Suguru Ueda, Yeoh Wen Liang, Osamu Fukuda, Kazuya Kikunaga.

**Methodology:** Tatsuya Honda, Kenshin Takemura, Susumu Matsumae.

**Project administration:** Shinya Ohmagari.

**Resources:** Shinya Ohmagari.

**Supervision:** Susumu Matsumae, Shinya Ohmagari.

**Writing – original draft:** Tatsuya Honda, Kenshin Takemura, Susumu Matsumae.

**Writing – review & editing:** Tatsuya Honda, Kenshin Takemura, Susumu Matsumae, Nobutomo Morita, Wataru Iwasaki, Ryoji Arita, Suguru Ueda, Yeoh Wen Liang, Osamu Fukuda, Kazuya Kikunaga, Shinya Ohmagari.

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
