## [Decision Letter · Decision Letter 0]

5 Nov 2023

PONE-D-23-33392Quantification of Caffeine in Coffee Cans Using Electrochemical Measurements, Machine Learning, and Boron-doped Diamond ElectrodesPLOS ONE

Dear Dr. Takemura,

Thank you for submitting your manuscript to PLOS ONE. After careful consideration, we feel that it has merit but does not fully meet PLOS ONE’s publication criteria as it currently stands. Therefore, we invite you to submit a revised version of the manuscript that addresses the points raised during the review process.

The major issues include the lack of proper statistical analysis, the absence of replicates, and the outdated estimation of the limit of detection. The experimental section is disorganized, and essential information is missing. The literature review should be more comprehensive, and a table summarizing relevant results is needed. The use of boron-doped diamond electrodes and their comparison with existing literature should be elaborated. The discussion section needs to be concise and focused. The choice of electrolyte and discrepancies in machine learning results require explanation. Proper comparisons with HPLC and clarification of the recovery procedure are needed. Addressing these concerns raised by reviewers will significantly enhance the manuscript's quality.

We look forward to receiving your revised manuscript.

Kind regards,

Pramod Kumar Gupta, Ph.D.

Academic Editor

PLOS ONE

2. Thank you for stating the following financial disclosure: " the Adaptable and Seamless Technology transfer Program through Target-driven R&D (A-STEP) from Japan Science and Technology Agency Grant Number JPMJTR22R2."

In addition to the reviewers' input, please consider the following points:

1. Elaborate on the film thickness and boron doping level within the Boron-Doped Diamond (BDD) material, as the electrochemical characteristics are known to significantly fluctuate with varying levels of these parameters.

2. Describe the pretreatment steps that were applied to the BDD material to ensure its stability and electrochemical activity.

3. Specify whether the BDD material used in your study was hydrogen-terminated (H-terminated) or hydroxyl-terminated (OH-terminated).

4. Provide detailed information regarding the environmental conditions maintained during the measurements in your experiments.

Reviewers' comments:

Reviewer's Responses to Questions

**Comments to the Author**

1. Is the manuscript technically sound, and do the data support the conclusions?

Reviewer #1: Yes

Reviewer #2: Partly

Reviewer #3: Yes

Reviewer #4: Partly

2. Has the statistical analysis been performed appropriately and rigorously? 

Reviewer #1: Yes

Reviewer #2: No

Reviewer #3: Yes

Reviewer #4: No

3. Have the authors made all data underlying the findings in their manuscript fully available?

Reviewer #1: Yes

Reviewer #2: No

Reviewer #3: Yes

Reviewer #4: No

4. Is the manuscript presented in an intelligible fashion and written in standard English?

Reviewer #1: Yes

Reviewer #2: Yes

Reviewer #3: Yes

Reviewer #4: Yes

5. Review Comments to the Author

Reviewer #1: In my opinion, the application of machine learning tool for current work provides an additional data in addition to the experimental results. This application is a very interesting part for the present work that provide significance to the current work. However, there are some major concerns that must be addressed by the authors in the electrochemical experiment parts. After this review the manuscript can be accpeted for the publication

Reviewer #2: Quantification of Caffeine in Coffee Cans Using Electrochemical Measurements, Machine Learning, and Boron-doped Diamond Electrodes. By applying PCA, they predict the caffeine content in unknown beverages. However, the work needs a wide revision of the literature regards to the electrochemical methods used to detect caffeine e more details of the experimental procedures and comparison of the results obtained with other reports in the literature and with reference methods. Thus, a major review is needed.

For introduction

Here: “Moreover, the redox voltages of the compounds that react at the electrode interface often overlap, making peak separation challenging. Boron doped diamond electrodes have excellent characteristics, such as a wide-potential window, low background current, and long-term response stability [14,15]. A wide-potential window contributes to various measurable redox voltages. Furthermore, changes have been observed in the current value at which the caffeine redox reaction occurs.”

1. There are several difficult to measure caffeine in several types of sample due to its high potential of oxidation. Example, there are several difficult to measure caffeine in presence of theobromine and theophylline due to the oxidations potential to be very near among them by using the BDD electrode. Please comment about this problem in your revision, and the limit detection obtained compared to your work. Moreover, please insert a table for comparison purposes to compare the linear range, LD, accuracy, precision, recovery data, work electrode, voltammetric method, type of sample, etc…. Please consult this and other rappers to make a table for comparison proposals.

1. Spãtaru at al. (Anodic voltammetry of xanthine, theophylline, theobromine and caffeine at conductive diamond electrodes and its analytical application) link: https://analyticalsciencejournals.onlinelibrary.wiley.com/doi/epdf/10.1002/1521-4109(200206)14:11%3C721::AID-ELAN721%3E3.0.CO;2-1?src=getftr

2. Gomes et al. (Doi 10.1016/j.foodcont.2019.106887)

3. Amare et al. https://doi.org/10.1155/2017/3979068

And other…..

2. The comparison with quantification using the proposed methods of an HPLC need to be performed, employing a F and T-test statistical tests for a confidence level.

3. The procedure to perform the recovery, with equations used should be more clear.

4. To oxidate caffeine, the faradaic peaks between 0.4-0.8 V and another peak at 1.6 V vs. ref. were observed. Please check if the peak at 0.4-0.8 V is with regards to the caffeine by comparison with the literature. Another problem is that the first peak at 0.4-0.8 V presented several interferences. Please comment about it.

Reviewer #3: In the current study, the authors quantified the caffeine content in different coffee cans using electrochemical measurements, boron-doped diamond electrodes, and machine learning techniques. The developed boron-doped diamond electrode was further characterized with SEM, AFM, and electrochemical stability. Substory, Asahi, KIRIN, COKA-COLA, and UCC were commercially available caffeine products used in the current study.

The following correction needs to be carried out.

1. COKA-COLA spelling may be verified again.

2. Suntory product A can be modified as a Suntory sample, if the author wants to change it.

3. In Table 2, labeling should be provided carefully to differentiate the samples, For Example, KIRIN product A or 1 and KIRIN product B or 2, and the same trends should be followed for other samples in Table 2.

Reviewer #4: The manuscript deserves publication after major revisions:

I consider that the statistical analysis is lacking (the method was partially evaluated) and according to results reported in the manuscript, clearly no replicates have been made. I consider that some data seem to be necessary to complete the work and improve its scientific quality such as error intervals, LOD, LOQ (indicating equations used), robustness, recovery with standard deviation, uncertainty of measurement, standard deviation, etc.

The limit of detection has been estimated using the old, now abandoned, IUPAC definition. Please adhere to modern standards, by using the accepted IUPAC recommendations based on types I and II errors (false positives and false negatives) quoted in L. A. Currie, Recommendations in Evaluation of Analytical Methods including Detection and Quantification Capabilities, Pure Appl. Chem. 1995, 67, 1699-1723.

The correlation coefficients of the calibration curves do not indicate a linear relationship between concentration and signal. Authors should report the experimental F value (and the tabulated critical F for comparison), as detailed in K. Danzer and L. A. Currie, Guidelines for calibration in analytical chemistry. Part 1. Fundamentals and single component calibration, Pure & Appl. Chem. 1998, 70, 993-1014. This F test is the best linearity indicator, as recommended by IUPAC. See the excellent brief report by the Analytical Methods Committee, No. 3, Dec 2000, from The Royal Society of Chemistry: Is my calibration linear? Available in http://www.rsc.org/pdf/amc/brief3.pdf.

In addition, authors should provide sufficient information to enable the reader to establish quickly and unambiguously the exact conditions used, and also to evaluate properly the experimental results such as stability constant measurements, principally, regarding on the caffeine detection.

Some phrases are finished without references, and I consider that this information is supported by other authors, then, please include the respective references citations.

Please, check some phrases that don not have sense in the text or that the intention of the authors were different to express specific information.

Experimental section is disorganized because I did not understand …when, what or how …the authors used the sensors to detect or quantify caffeine.

The main purpose of the work, as indicated by title, is the use of boron doped diamond electrode, however, no obvious data have been cited where the diamond electrode was previously used by other authors. In fact, there are several examples in the literature. Additionally, caffeine was also determined by using innovative electrochemical sensors, then, the authors should construct a table containing the most important results reported in the existing literature.

I consider that the response of other chemical species could be tested with these sensors in order to determine the preference species (positive or negative species), permeability, redox rate, and other kinetic parameters. In addition, depending on the preference species, these sensors could be used to quantify or identify organic compounds; therefore, I consider that the authors could be more ambitious to explore supplementary characteristics of this kind of electrodes.

Characterization of diamond electrodes is well-known, then, this info is not needed. It is not new.

How the figures of merit were estimated, should be indicated.

Discussion is superfluous then it should be improved.

The use of NaCl is not a correct selection because diamond electrodes at potentials higher than 1.2V can promote the production of active chlorine species, and at the caffeine potential, these species are available to promote the caffeine oxidation. Then, how it is avoided? How it is benefit? Selection of other electrolyte is needed to compare with the results obtained.

6. PLOS authors have the option to publish the peer review history of their article (what does this mean?). If published, this will include your full peer review and any attached files.

Reviewer #1: **Yes: **Dr. Deepika Chauhan

Reviewer #2: **Yes: **Vagner Bezerra dos Santos

Reviewer #3: No

Reviewer #4: No

---

## [Author Response · Author response to Decision Letter 0]

11 Dec 2023

5. Review Comments to the Author

We thank the Reviewers for their thoughtful suggestions and insights, which have enriched the manuscript and produced a better and more balanced account of the research.

Reviewer #1: In my opinion, the application of machine learning tool for current work provides an additional data in addition to the experimental results. This application is a very interesting part for the present work that provide significance to the current work. However, there are some major concerns that must be addressed by the authors in the electrochemical experiment parts. After this review the manuscript can be accpeted for the publication.

Response: Based on your suggestions, we have included the measurement data of 10 samples in actual coffee cans as shown in Fig 4. Each sample was measured by replacing the solution thrice and replacing the electrode each time. The revised and newly included graphs are listed below. 

Page 17, line 269-272

Ten samples from five companies were assayed, including a sample of decaffeinated coffee with 60 mg of caffeine added (Fig 4). Table 1 presents the quantification results of the study. Each sample was measured three times and subjected to quantification by machine learning (S3 Fig).

Fig 4. Results of SWV measurements of 10 coffee samples from five companies.

S3 Fig. A-I) Results of 10 coffee samples from 5 companies measured repeatedly by SWV.

Reviewer #2: Quantification of Caffeine in Coffee Cans Using Electrochemical Measurements, Machine Learning, and Boron-doped Diamond Electrodes. By applying PCA, they predict the caffeine content in unknown beverages. However, the work needs a wide revision of the literature regards to the electrochemical methods used to detect caffeine e more details of the experimental procedures and comparison of the results obtained with other reports in the literature and with reference methods. Thus, a major review is needed.

For introduction

Here: “Moreover, the redox voltages of the compounds that react at the electrode interface often overlap, making peak separation challenging. Boron doped diamond electrodes have excellent characteristics, such as a wide-potential window, low background current, and long-term response stability [14,15]. A wide-potential window contributes to various measurable redox voltages. Furthermore, changes have been observed in the current value at which the caffeine redox reaction occurs.”

1. There are several difficult to measure caffeine in several types of sample due to its high potential of oxidation. Example, there are several difficult to measure caffeine in presence of theobromine and theophylline due to the oxidations potential to be very near among them by using the BDD electrode. Please comment about this problem in your revision, and the limit detection obtained compared to your work. Moreover, please insert a table for comparison purposes to compare the linear range, LD, accuracy, precision, recovery data, work electrode, voltammetric method, type of sample, etc…. Please consult this and other rappers to make a table for comparison proposals.

1. Spãtaru at al. (Anodic voltammetry of xanthine, theophylline, theobromine and caffeine at conductive diamond electrodes and its analytical application) link: https://analyticalsciencejournals.onlinelibrary.wiley.com/doi/epdf/10.1002/1521-4109(200206)14:11%3C721::AID-ELAN721%3E3.0.CO;2-1?src=getftr

2. Gomes et al. (Doi 10.1016/j.foodcont.2019.106887)

3. Amare et al. https://doi.org/10.1155/2017/3979068

And other…..

Response: We thank the Reviewer for pointing this out to us. We are aware of this limitation inherent in our study and have tried to imply it in the study's title, emphasizing its focus specifically on coffee cans. The major advantage of our research is that we have succeeded in estimating the amount of caffeine from undiluted coffee cans, which contain a significant amount of foreign substances and are easily ridden by foreign signals, without any preprocessing by using PCA and machine learning. The HPLC molecular analysis also shows that theobromine and theophylline are not present in the coffee can. Moreover, in comparison with other reported cases, the coffee solution content, which is the solvent used in the measurement, differs significantly with the manufacturer. Therefore, it is challenging to compare the results without standardizing the solvent conditions by mixing them with a specific electrolyte or acid solution, as in other reported cases where high-quality electrochemical measurements were performed. We agree that the points you have raised are critical to the development of this research and must be resolved to extend its versatility to other substances. Therefore, we have acknowledged the same in the conclusion. The revisions made to the conclusion are as follows. 

Page 26, line 334-339

The combination of electrochemical measurements and machine learning showed that quantitative estimation is possible even from solutions containing foreign substances, as long as the peaks are prominent. However, many organic compounds, including caffeine, have similar oxidation potentials. To make this technique more versatile, a new data learning method that mechanically processes the minor differences in oxidation potential between substances with similar oxidation potentials to separate peaks at a high level is required.

Response: Regarding comparisons with other papers, none of the prior literature was a suitable comparison, as the electrochemical measurement conditions were optimized in all of them. We did not create a table because the content raised concerns that it would reduce the resolution to the reader. Please point this out again if it is insufficient.

2. The comparison with quantification using the proposed methods of an HPLC need to be performed, employing a F and T-test statistical tests for a confidence level.

Response: A quantitative analysis was performed on the samples processed for HPLC with the help of the Food Cosmetics Department of the Saga Prefectural Industrial Technology Center, which has experience in the HPLC measurement of beverages. The quantitative results demonstrated that the caffeine content in all samples was lower than the amount stated on the label. There is a possibility of loss due to the pretreatment. Since comparison with HPLC is not an appropriate method when the amount listed by the manufacturer is used as the reference value, as in this case, the amount was not added to the mass spectrometry (MS). 

Table: Quantification of the caffeine amount in a coffee can using HPLC-MS 

Sample No. Manufacture

(published caffeine value mg/100 mg) Containing Caffeine amount, mg/100 mL

 Mean SD

1 Asahi sample A

(70 mg/100 mg) Milk, coffee, sugar, whole milk powder, dextrin, emulsifier, casein sodium, flavor, VC, acesulfame K, sucralose 41.4 0.1

2 Asahi sample A

(80 mg/100 mg) Milk, coffee, sugar, skimmed milk powder, processed cream, xylitol, acesulfame K, sucralose, emulsifier, casein sodium 41.6 0.2

3 Kirin sample A

(80 mg/100 mg) Coffee, flavoring 32.3 0.1

4 Suntory sample A

(40 mg/100 mg) Coffee 27.2 0.0

3. The procedure to perform the recovery, with equations used should be more clear.

Response: Acknowledging your suggestion, the formula used for the calculation has been included in the manuscript as follows.

Page 10, line 187

A more concise equation for deriving the quantitative values from the output data was calculated following equation. α, β, γ : median value μ: average value of median values. 

α=(X_1+Y_1 ),β=(X_2+Y_2 ),γ=(X_3+Y_3 )

μ=ε÷2

The obtained results were equivalent to this equation.

4. To oxidate caffeine, the faradaic peaks between 0.4-0.8 V and another peak at 1.6 V vs. ref. were observed. Please check if the peak at 0.4-0.8 V is with regards to the caffeine by comparison with the literature. Another problem is that the first peak at 0.4-0.8 V presented several interferences. Please comment about it.

Response: There have been no reported cases of increased current values owing to the oxidation of caffeine at the potentials mentioned in the literature. In contrast, caffeine may exhibit different electrochemical behaviors when reacting with salts. We believe that the peak in the graph you pointed out was caused by the suspension of caffeine in a significant amount of salt. Moreover, the peaks between 0.4 and 0.8 V are the oxidation potentials of large amounts of organic compounds and hydroxy groups, as you rightly pointed out, which can cause a lot of interference. In the actual coffee can, a strong increase in current was observed in the corresponding area, but no such interference was observed around 1.6 V. This method enables quantification using the 1.6 V caffeine peak from machine learning, even if the undiluted solution is used for electrochemical measurement without any pretreatment. Therefore, the manufacturer's donation value can be quantified without being significantly affected by interference. We have included graphs of all measurements taken with actual undiluted coffee cans. Kindly find them below. 

Fig 4. Results of SWV measurements of 10 coffee samples from five companies.

S3 Fig. A-I) Results of 10 coffee samples from 5 companies measured repeatedly by SWV.

Reviewer #3: In the current study, the authors quantified the caffeine content in different coffee cans using electrochemical measurements, boron-doped diamond electrodes, and machine learning techniques. The developed boron-doped diamond electrode was further characterized with SEM, AFM, and electrochemical stability. Substory, Asahi, KIRIN, COKA-COLA, and UCC were commercially available caffeine products used in the current study.

The following correction needs to be carried out.

1. COKA-COLA spelling may be verified again.

2. Suntory product A can be modified as a Suntory sample, if the author wants to change it.

3. In Table 2, labeling should be provided carefully to differentiate the samples, For Example, KIRIN product A or 1 and KIRIN product B or 2, and the same trends should be followed for other samples in Table 2.

Response: Based on your suggestions, we have incorporated all changes in our manuscript. For your reference, as an example, Table 1 is listed below. 

Table 1 Caffeine quantification results for each algorithm.

Caffeine content 

(mg/100 g) 

Unknown solution Manufacturer 

published value Analysis 

evaluation Logical 

evaluation Graded 

evaluation

Suntory sample A-1 40 47.46 47.70 47.58

Suntory sample A-2 47.30 49.79 48.55

Suntory sample A-3 38.97 44.75 41.86

Suntory sample B-1 40 38.56 46.07 42.32

Suntory sample B-2 38.48 45.05 41.91

Suntory sample B-3 38.61 45.89 42.25

Coca-Cola sample A-1 60 62.48 62.94 62.71

Coca-Cola sample A-2 56.87 54.74 55.80

Coca-Cola sample A-3 62.72 66.57 64.65

UCC sample A-1

(Decaffeinated coffee)

+anhydrous caffeine 60 50.44 52.85 51.65

UCC sample A-2

(Decaffeinated coffee)

+ anhydrous caffeine 57.14 55.36 56.25

UCC sample A-3

(Decaffeinated coffee)

+ anhydrous caffeine 49.19 51.11 50.15

Asahi sample A-1 62 59.03 54.43 56.73

Asahi sample A-2 60.89 60.19 60.54

Asahi sample A-3 63.70 65.10 64.40

KIRIN sample A-1 70 70.00 76.72 73.36

KIRIN sample A-2 64.14 70.80 67.47

KIRIN sample A-3 61.33 63.04 62.18

Asahi sample B-1 70 76.34 74.76 75.55

Asahi sample B-2 70.78 66.59 68.69

Asahi sample B-3 70.5 65.69 68.09

KIRIN sample B-1 78 75.05 64.24 69.64

KIRIN sample B-2 75.69 70.79 73.24

KIRIN sample B-3 76.18 71.89 74.03

Asahi sample C-1 80 80.78 85.60 83.19

Asahi sample C-2 83.02 80.42 81.72

Asahi sample C-3 82.65 77.42 80.04

Suntory sample C-1 82 78.27 82.47 80.37

Suntory sample C-2 81.90 91.80 86.85

Suntory sample C-3 78.51 77.19 77.85

Reviewer #4: The manuscript deserves publication after major revisions:

I consider that the statistical analysis is lacking (the method was partially evaluated) and according to results reported in the manuscript, clearly no replicates have been made. I consider that some data seem to be necessary to complete the work and improve its scientific quality such as error intervals, LOD, LOQ (indicating equations used), robustness, recovery with standard deviation, uncertainty of measurement, standard deviation, etc.

The limit of detection has been estimated using the old, now abandoned, IUPAC definition. Please adhere to modern standards, by using the accepted IUPAC recommendations based on types I and II errors (false positives and false negatives) quoted in L. A. Currie, Recommendations in Evaluation of Analytical Methods including Detection and Quantification Capabilities, Pure Appl. Chem. 1995, 67, 1699-1723.

The correlation coefficients of the calibration curves do not indicate a linear relationship between concentration and signal. Authors should report the experimental F value (and the tabulated critical F for comparison), as detailed in K. Danzer and L. A. Currie, Guidelines for calibration in analytical chemistry. Part 1. Fundamentals and single component calibration, Pure & Appl. Chem. 1998, 70, 993-1014. This F test is the best linearity indicator, as recommended by IUPAC. See the excellent brief report by the Analytical Methods Committee, No. 3, Dec 2000, from The Royal Society of Chemistry: Is my calibration linear? Available in http://www.rsc.org/pdf/amc/brief3.pdf.

Response: We thank you for your valuable feedback. Kindly find our explanation for the same.

Actual sample evaluations of caffeine detection in coffee cans by five different companies' products revealed significant differences in content as well. The purpose of this study was to estimate the amount of caffeine without any processing, such as mixing an undiluted solution with a specific solvent. Therefore, it was not possible to mix a specific electrolyte mass with caffeine to stabilize caffeine kinetics, as in other reported cases of electrochemical detection of caffeine. Instead of preparing high-quality measurement data by constructing the conditions necessary to determine the LoD and LoQ, machine learning and PCA were used to estimate caffeine content from graphs that were challenging to estimate by the human eye at first glance. The emphasis of this study was to use PCA to quantify the amount of caffeine.

In addition, authors should provide sufficient information to enable the reader to establish quickly and unambiguously the exact conditions used, and also to evaluate properly the experimental results such as stability constant measurements, principally, regarding on the caffeine detection.

Reply: The relevant information was included in the experimental procedure to help any future reproduction of the experiment. The revision is as follows.

Page 7, line 113-174

Electrochemical measurement of caffeine in solution. 

Experiments were conducted to measure caffeine levels in commercial coffee without pretreatment. The pH levels of five different commercial coffee beverages were obtained. To verify the electrochemical response to caffeine, square wave voltammetry (SWV) measurements were performed using a 1 g/L NaCl solution containing 60 mg/100 g of anhydrous caffeine. The measurements were obtained from -2.4 V to 2.5 V in 0.016 V steps, with a 2 V voltage applied for 100 s to clean the electrode. A 3 ml solution was taken from the coffee can and directly injected into the measuring cell to perform the caffeine measurement by SWV. Absorbance measurements were performed from 220 to 800 nm to check for contamination at the electrode interface using UV-2600i (Shimadzu Co. Ltd., Kyoto, Japan).

Principal component analysis and machine learning algorism for caffeine quantification. 

As shown in Fig 1A, the electrochemically measured solution data were stored in the training data file, and the data for one of the solutions (one solution was measured three times) was stored as an unknown solution in the test data file. The solution data stored in the training data file was used to create a regression dataset of unknown solutions. The manufacturer's published value was provided as the correct solution data for each solution. The solution data stored in the test data file underwent principal component regression according to the data frame created by the data stored in the training data file. For this, principal component analysis, normalization, and grid research were performed on the solution data. Principal component analysis (PCA) was performed using Figs 1B and 1C. In this study, we developed an algorithm that combines an analytical evaluation value (the value extracted from the whole part where the feature value is strongly expressed), as shown in Fig 1B, which is the coordinate with good results of PCA for the solution data, and a logical evaluation value (value using the area of the narrow part where the redox reaction of caffeine occurs) (Fig 1C), which is the area around the peak voltage of 1.6 V where the oxidation reaction of caffeine occurs [22]. Therefore, the data entered in the training data file creates two data sets based on these coordinates and areas. Principal component regression (PCR) [23] was performed on the solution data stored in the test data file according to the two created data sets to obtain predicted values from each. The algorithm shown in Fig 1B can obtain a value close to the caffeine content. As shown in Fig 1B, predicted values (X1–X3) were obtained using only concrete information. The algorithm shown in Fig 1C uses an area divided into 17 segments because 51 coordinates exist between 1.55 V and 1.65 V. Predictions were made using an abstract feature from the overall information (the details alone from the entire area over which the caffeine oxidation reaction occurred). Using this method, predicted values (Y1–Y3) are obtained from the area. These predicting algorithms had low accuracy when used individually. Therefore, combining these two algorithms improves the average accuracy of the predicted values by evenly following electrochemical measurement data, including measurement errors owing to each electrode.

Following is the explanation of the process until the data stored in the test data file are quantified. The PCA obtained in Figs 1B and 1C is used to perform the same analysis for the unknown solution. Specifically, the solution data stored in the test data file is also subjected to the principal component analysis and normalization used when creating the data set. The data stored in the test file is adapted to the dataset. PCR was used to predict the unknown solutions from the variance obtained from the PCA results [24]. Because each solution was measured three times, three specific predictions (X1–X3) for solution A were obtained from Fig 1B, and three abstract predictions (Y1–Y3) were obtained from Fig 1C. The predicted values were averaged at each step of the hierarchy, as shown in Fig 1D. These values were considered the average values (Ave1–Ave3). The median of the stepwise evaluation values obtained becomes a predicted value that is finally output as a quantitative value, as shown in Fig 1E. 

A more concise equation for deriving the quantitative values from the output data was calculated following equation. α, β, γ : median value μ: average value of median values. 

α=(X_1+Y_1 ),β=(X_2+Y_2 ),γ=(X_3+Y_3 )

μ=ε÷2

The obtained results were equivalent to this equation.

Some phrases are finished without references, and I consider that this information is supported by other authors, then, please include the respective references citations.

Please, check some phrases that don not have sense in the text or that the intention of the authors were different to express specific information.

Experimental section is disorganized because I did not understand …when, what or how …the authors used the sensors to detect or quantify caffeine.

The main purpose of the work, as indicated by title, is the use of boron doped diamond electrode, however, no obvious data have been cited where the diamond electrode was previously used by other authors. In fact, there are several examples in the literature. 

Response: More details were made to the experimental techniques. Revisions, in addition to those mentioned in the previous section, were also made.

Additionally, caffeine was also determined by using innovative electrochemical sensors, then, the authors should construct a table containing the most important results reported in the existing literature.

I consider that the response of other chemical species could be tested with these sensors in order to determine the preference species (positive or negative species), permeability, redox rate, and other kinetic parameters. In addition, depending on the preference species, these sensors could be used to quantify or identify organic compounds; therefore, I consider that the authors could be more ambitious to explore supplementary characteristics of this kind of electrodes.

Response: This method has the advantage of being a quantitative method that mechanically processes electrochemical measurement data under high interference from foreign signals.　Conversely, if the composition of the solvent changes significantly, the statistical processing and study methods will need to be reconfigured. This study focused on the caffeine in coffee cans, which has relatively low peak adulteration with various organic substances, and succeeded in quantifying the values provided by the manufacturer. For multiple electrochemical analyses, it is necessary to set more rigorous electrochemical measurement parameters, select electrode materials, and devise new mechanical processing methods. This has been planned as the future scope of our research.

Characterization of diamond electrodes is well-known, then, this info is not needed. It is not new.

How the figures of merit were estimated, should be indicated.

Discussion is superfluous then it should be improved.

Response: The text has been revised to address the need for discussion. The interfacial and electrochemical analysis of diamond electrodes shows data that are not yet available for polycrystalline electrodes fabricated by the special CVD fabrication method used in this study. Therefore, we believe that the inclusion of these data in the manuscript is relevant.

Regarding the discussion part, electrochemical measurement data of 10 coffee can samples were newly included. We believe that this addition brings clarity to our discussion.

Ten samples from five companies were assayed, including a sample of decaffeinated coffee with 60 mg of caffeine added (Fig 4). Table 1 presents the quantification results of the study. Each sample was measured three times and subjected to quantification by machine learning (S3 Fig).

Fig 4. SWV measurement results of 10 coffee samples from five companies.

The use of NaCl is not a correct selection because diamond electrodes at potentials higher than 1.2V can promote the production of active chlorine species, and at the caffeine potential, these species are available to promote the caffeine oxidation. Then, how it is avoided? How it is benefit? Selection of other electrolyte is needed to compare with the results obtained.

Response: This study used machine learning and PCA to quantify caffeine without pretreatment or mixing with solvents, even in the presence of large amounts of foreign substances. Therefore, in anticipation of direct measurement from the undiluted solution, NaCl, which is not a suitable condition, was used as the solvent. We believe that this method has the advantage of allowing the estimation of caffeine content in any solvent, which is challenging, without aiming to make electrochemically high quality measurements.

S1 Table. pH measurements of beverages. 

Name pH salt equivalent amount (g/L)

Caffeine in 1 g/L NaCl 7.45 1

Suntory product 6.50 1.1

KIRIN product 6.08 0.4

Suntory product 6.05 1

Georgia product 6.54 1

KIRIN product 6.54 0.7

Asahi product 6.42 0.5

Suntory product 6.67 1

---

## [Decision Letter · Decision Letter 1]

10 Jan 2024

PONE-D-23-33392R1Quantification of Caffeine in Coffee Cans Using Electrochemical Measurements, Machine Learning, and Boron-doped Diamond ElectrodesPLOS ONE

Dear Dr. Takemura,

Thank you for submitting your manuscript to PLOS ONE. After careful consideration, we feel that it has merit but does not fully meet PLOS ONE’s publication criteria as it currently stands. Therefore, we invite you to submit a revised version of the manuscript that addresses the points raised during the review process.

We look forward to receiving your revised manuscript.

Kind regards,

Pramod Kumar Gupta, Ph.D.

Academic Editor

PLOS ONE

Journal Requirements:

**Additional Editor Comments>**:

**In addition to the reviewers' input, please consider the following points while revising the manuscript**:

**1. Elaborate on the film thickness and boron doping level within the Boron-Doped Diamond (BDD) material, as the electrochemical characteristics are known to significantly fluctuate with varying levels of these parameters**.

**2. Describe the pretreatment steps that were applied to the BDD material to ensure its stability and electrochemical activity**.

**3. Specify whether the BDD material used in your study was hydrogen-terminated (H-terminated) or hydroxyl-terminated (OH-terminated)**.

**4. Provide detailed information regarding the environmental conditions maintained during the measurements in your experiments.**

Reviewers' comments:

Reviewer's Responses to Questions

**Comments to the Author**

1. If the authors have adequately addressed your comments raised in a previous round of review and you feel that this manuscript is now acceptable for publication, you may indicate that here to bypass the “Comments to the Author” section, enter your conflict of interest statement in the “Confidential to Editor” section, and submit your "Accept" recommendation.

Reviewer #2: (No Response)

Reviewer #3: All comments have been addressed

Reviewer #4: All comments have been addressed

2. Is the manuscript technically sound, and do the data support the conclusions?

Reviewer #2: Partly

Reviewer #3: Yes

Reviewer #4: Yes

3. Has the statistical analysis been performed appropriately and rigorously? 

Reviewer #2: No

Reviewer #3: N/A

Reviewer #4: Yes

4. Have the authors made all data underlying the findings in their manuscript fully available?

Reviewer #2: No

Reviewer #3: Yes

Reviewer #4: Yes

5. Is the manuscript presented in an intelligible fashion and written in standard English?

Reviewer #2: Yes

Reviewer #3: Yes

Reviewer #4: Yes

6. Review Comments to the Author

Reviewer #2: The authors answered partially the questions, and a complete response is required. All these comments should be inserted in the revised manuscript, because these comments are important to the readers. Some few were inserted in the manuscript, but all suggestions need to be used. The introduction with suggested references and others also is needed. Thus, a minor revision is required. New review is marked as “Review-R2”.

Question 1:

For introduction Here: “Moreover, the redox voltages of the compounds that react at the electrode interface often overlap, making peak separation challenging. Boron doped diamond electrodes have excellent characteristics, such as a wide-potential window, low background current, and long-term response stability [14,15]. A wide-potential window contributes to various measurable redox voltages. Furthermore, changes have been observed in the current value at which the caffeine redox reaction occurs.” 1. There are several difficult to measure caffeine in several types of sample due to its high potential of oxidation. Example, there are several difficult to measure caffeine in presence of theobromine and theophylline due to the oxidations potential to be very near among them by using the BDD electrode. Please comment about this problem in your revision, and the limit detection obtained compared to your work. Moreover, please insert a table for comparison purposes to compare the linear range, LD, accuracy, precision, recovery data, work electrode, voltammetric method, type of sample, etc…. Please consult this and other rappers to make a table for comparison proposals. 1. Spãtaru at al. (Anodic voltammetry of xanthine, theophylline, theobromine and caffeine at conductive diamond electrodes and its analytical application) link: https://analyticalsciencejournals.onlinelibrary.wiley.com/doi/epdf/10.1002/1521- 4109(200206)14:11%3C721::AID-ELAN721%3E3.0.CO;2-1?src=getftr 2. Gomes et al. (Doi 10.1016/j.foodcont.2019.106887) 3. Amare et al. https://doi.org/10.1155/2017/3979068 And other…..

Response: We thank the Reviewer for pointing this out to us. We are aware of this limitation inherent in our study and have tried to imply it in the study's title, emphasizing its focus specifically on coffee cans. The major advantage of our research is that we have succeeded in estimating the amount of caffeine from undiluted coffee cans, which Powered by Editorial Manager® and ProduXion Manager® from Aries Systems Corporation contain a significant amount of foreign substances and are easily ridden by foreign signals, without any preprocessing by using PCA and machine learning. The HPLC molecular analysis also shows that theobromine and theophylline are not present in the coffee can. Moreover, in comparison with other reported cases, the coffee solution content, which is the solvent used in the measurement, differs significantly with the manufacturer. Therefore, it is challenging to compare the results without standardizing the solvent conditions by mixing them with a specific electrolyte or acid solution, as in other reported cases where high-quality electrochemical measurements were performed. We agree that the points you have raised are critical to the development of this research and must be resolved to extend its versatility to other substances. Therefore, we have acknowledged the same in the conclusion. The revisions made to the conclusion are as follows. Page 26, line 334-339 The combination of electrochemical measurements and machine learning showed that quantitative estimation is possible even from solutions containing foreign substances, as long as the peaks are prominent. However, many organic compounds, including caffeine, have similar oxidation potentials. To make this technique more versatile, a new data learning method that mechanically processes the minor differences in oxidation potential between substances with similar oxidation potentials to separate peaks at a high level is required. Response: Regarding comparisons with other papers, none of the prior literature was a suitable comparison, as the electrochemical measurement conditions were optimized in all of them. We did not create a table because the content raised concerns that it would reduce the resolution to the reader. Please point this out again if it is insufficient.

Review-R2: A Table for comparison of data or features, highlighting is possible being optimized or not. Thus, please insert it in the revised manuscript.

Review-R2: The potential interferent compounds are a problem, thus, the electroactive compound with oxidation potential leads to an accurate data. Thus, it is important to comment on it. The only comment in the conclusion is not sufficient, because the article and its discussions need to be inserted in the revised manuscript.

Review-R2: A comparison table is possible to be constructed and to show the main highlights of the manuscript compared to others, to measure caffeine for coffee and other samples to evaluate the performance of the electrochemical methods and the method proposed here. For quantitative purposes, the limit of detection and quantification, linear range are always supplied, thus, the authors should be presented with a comparison of the method reported in the literature and the proposed method.

2. The comparison with quantification using the proposed methods of an HPLC need to be performed, employing a F and T-test statistical tests for a confidence level. Response: A quantitative analysis was performed on the samples processed for HPLC with the help of the Food Cosmetics Department of the Saga Prefectural Industrial Technology Center, which has experience in the HPLC measurement of beverages. The quantitative results demonstrated that the caffeine content in all samples was lower than the amount stated on the label. There is a possibility of loss due to the pretreatment. Since comparison with HPLC is not an appropriate method when the amount listed by the manufacturer is used as the reference value, as in this case, the amount was not added to the mass spectrometry (MS). Table: Quantification of the caffeine amount in a coffee can using HPLC-MS Sample No.Manufacture (published caffeine value mg/100 mg)Containing Caffeine amount, mg/100 mL MeanSD 1Asahi sample A (70 mg/100 mg)Milk, coffee, sugar, whole milk powder, dextrin, emulsifier, casein sodium, flavor, VC, acesulfame K, sucralose41.40.1 2Asahi sample A (80 mg/100 mg)Milk, coffee, sugar, skimmed milk powder, processed cream, xylitol, acesulfame K, sucralose, emulsifier, casein sodium41.60.2 3Kirin sample A (80 mg/100 mg)Coffee, flavoring32.30.1 4Suntory sample A (40 mg/100 mg)Coffee27.20.0

Review-R2: Ok, but a statistical test should be performed. Please insert the data and prove this possible inconsistency.

3. The procedure to perform the recovery, with equations used should be more clear. Response: Acknowledging your suggestion, the formula used for the calculation has been included in the manuscript as follows. Page 10, line 187 A more concise equation for deriving the quantitative values from the output data was calculated following equation. α, β, γ : median value μ: average value of median values. α=(X_1+Y_1 ),β=(X_2+Y_2 ),γ=(X_3+Y_3 ) Powered by Editorial Manager® and ProduXion Manager® from Aries Systems Corporation μ=ε÷2 The obtained results were equivalent to this equation.

Review-R2: Why were the recovery tests always lower than 100%?

4. To oxidate caffeine, the faradaic peaks between 0.4-0.8 V and another peak at 1.6 V vs. ref. were observed. Please check if the peak at 0.4-0.8 V is with regards to the caffeine by comparison with the literature. Another problem is that the first peak at 0.4- 0.8 V presented several interferences. Please comment about it.

Response: There have been no reported cases of increased current values owing to the oxidation of caffeine at the potentials mentioned in the literature. In contrast, caffeine may exhibit different electrochemical behaviors when reacting with salts. We believe that the peak in the graph you pointed out was caused by the suspension of caffeine in a significant amount of salt. Moreover, the peaks between 0.4 and 0.8 V are the oxidation potentials of large amounts of organic compounds and hydroxy groups, as you rightly pointed out, which can cause a lot of interference. In the actual coffee can, a strong increase in current was observed in the corresponding area, but no such interference was observed around 1.6 V. This method enables quantification using the 1.6 V caffeine peak from machine learning, even if the undiluted solution is used for electrochemical measurement without any pretreatment. Therefore, the manufacturer's donation value can be quantified without being significantly affected by interference. We have included graphs of all measurements taken with actual undiluted coffee cans. Kindly find them below. Fig 4. Results of SWV measurements of 10 coffee samples from five companies. S3 Fig.

Review-R2: This comment needs to be inserted in the manuscript, because 0.4-0.8 V is not used for Caffeine quantification, only the oxidation peak at 1.6 V. This high potential is easily found for other compounds; this information needs to be discussed in the revised manuscript.

Reviewer #3: Authors duly revised the manuscript according to the comments. Hence, the manuscript can be acceptable in the present form. Thank you.

Reviewer #4: All concerns have been carefully addressed and the manuscript is suitable for publication. The authors have extend the discussion section and give all parameters used in the experimental section in detail. Several figures have been improved. Other references have been added even when other reported already published have been not included for example, Martinez-Huitle group.

7. PLOS authors have the option to publish the peer review history of their article (what does this mean?). If published, this will include your full peer review and any attached files.

Reviewer #2: **Yes: **Vagner Bezerra dos Santos

Reviewer #3: No

Reviewer #4: No

---

## [Author Response · Author response to Decision Letter 1]

11 Jan 2024

Points pointed out by the editor are highlighted in yellow to indicate revisions. Changes made in response to the reviewers are highlighted in green.

---

## [Decision Letter · Decision Letter 2]

23 Jan 2024

Quantification of Caffeine in Coffee Cans Using Electrochemical Measurements, Machine Learning, and Boron-doped Diamond Electrodes

PONE-D-23-33392R2

Dear Dr. Takemura, 

We’re pleased to inform you that your manuscript has been judged scientifically suitable for publication and will be formally accepted for publication once it meets all outstanding technical requirements.

Kind regards,

Pramod Kumar Gupta, Ph.D.

Academic Editor

PLOS ONE

---

## [Editor Report · Acceptance letter]

15 Mar 2024

PONE-D-23-33392R2 

PLOS ONE

Dear Dr. Takemura, 

I'm pleased to inform you that your manuscript has been deemed suitable for publication in PLOS ONE. Congratulations! Your manuscript is now being handed over to our production team.

Kind regards, 

on behalf of

Dr. Pramod Kumar Gupta 

Academic Editor

PLOS ONE